# Impact of Body Mass Index on Outcomes in Pediatric Allogeneic Hematopoietic Stem Cell Transplantation Recipients: A Single-Center Retrospective Study

**DOI:** 10.3390/nu16213638

**Published:** 2024-10-25

**Authors:** Stefania Braidotti, Debora Curci, Davide Zanon, Alessandra Maestro, Antonella Longo, Nicole De Vita, Natalia Maximova

**Affiliations:** 1Department of Pediatrics, Institute for Maternal and Child Health-IRCCS Burlo Garofolo, 34137 Trieste, Italy; stefania.braidotti@burlo.trieste.it (S.B.); antonella.longo@burlo.trieste.it (A.L.); nicole.devita@burlo.trieste.it (N.D.V.); 2Advanced Translational Diagnostic Laboratory, Institute for Maternal and Child Health-IRCCS Burlo Garofolo, 34137 Trieste, Italy; debora.curci@burlo.trieste.it; 3Pharmacy and Clinical Pharmacology Department, Institute for Maternal and Child Health-IRCCS Burlo Garofolo, 34137 Trieste, Italy; davide.zanon@burlo.trieste.it (D.Z.); alessandra.maestro@burlo.trieste.it (A.M.)

**Keywords:** BMI, allo-HSCT, pediatric, nutritional status, transplant outcomes, supportive care

## Abstract

Background: Pediatric patients undergoing allogeneic hematopoietic stem cell transplantation (allo-HSCT) face several risk factors influencing transplantation success, including nutritional status as measured by body mass index (BMI). Methods: This study analyzed BMI data collected from patients transplanted between 2003 and 2023, and aimed to evaluate whether deviations from normal BMI are associated with poorer clinical outcomes. BMI levels assessed before and after first-line treatment and pre-transplantation were analyzed retrospectively to determine a correlation with survival and post-transplant complications. Results: Underweight patients had significantly lower 12- and 36-month overall survival rates compared to normal-weight and overweight patients (*p* = 1.22 × 10^−8^ and *p* = 8.88 × 10^−8^, respectively). Event-free survival was also lower for underweight patients at all time points. A higher pre-transplant BMI increases the risk of acute graft-versus-host disease (GVHD, *p* = 0.00068). Otherwise, pre-transplant BMI was not significantly correlated with early TRCs and cGVHD. As secondary objectives, this study identified differences in BMI across primary disease groups, with solid tumor patients having the highest BMI and myelodysplastic syndrome patients having the lowest. BMI cut-offs were identified to predict or protect against serious outcomes, including delayed engraftment, TRCs, and acute and chronic GVHD. Conclusions: This study highlights the importance of nutritional assessment and management in pediatric patients undergoing allo-HSCT to optimize post-transplant outcomes, as deviations from a normal BMI can significantly impact post-transplant health. These findings underscore the importance of integrating BMI assessment throughout the entire pre-HSCT therapeutic course to identify patients at higher risk for complications and to define more effective nutritional management strategies.

## 1. Introduction

Allogeneic hematopoietic stem cell transplantation (allo-HSCT) represents a remarkably effective and potentially life-saving treatment for hematological disorders and malignancies in pediatric patients. Allo-HSCT replaces defective or dysregulated recipient immune and hematopoietic cells with steady repopulating cells from a healthy donor [1,2]. However, the pre-transplant risk factors may affect long-term survival and quality of life. Despite advances in immunosuppressive therapy and complication management, allo-HSCT remains a complex procedure associated with significant mortality and morbidity, mainly related to the occurrence of a high risk of conditioning regimen toxicity and transplant-related complications (TRCs) [3,4]. Early and late TRCs can arise, including graft-versus-host disease (GVHD) and complications related to endothelial injury and infections [5]. Even disease recurrence and organ dysfunction are some of the major complications that can compromise long-term survival and quality of life of patients [6].

In recent years, nutritional status and individualizing nutritional support have also gained increasing attention. Body mass index (BMI), calculated as weight divided by height squared, is an anthropometric parameter widely used to assess nutritional status and has emerged as a potential prognostic factor in several diseases, including transplant procedures [7,8]. The high risk of malnutrition, intended as undernutrition, is a significant and growing issue in the pediatric population. It mainly affects patients who have undergone radio/chemotherapies to treat malignancies and undergone conditioning regimens for allo-HSCT procedures (almost 10–15%) [9]. Additionally, multiple chemotherapy treatments could lead to the insurance of adverse effects, among which oral and/or enteral mucositis and other gastrointestinal sequelae, such as vomiting, diarrhea, and anorexia [10]. It is reasonable to consider that an insufficient or excessive BMI could negatively impact transplant outcomes, potentially through mechanisms such as inflammation, oxidative stress, and immune dysfunction. In addition, BMI might reflect an impaired nutritional status, which may affect the tolerance to myelosuppressive therapy and the ability to recover after transplantation.

Nutritional status assessment at the time of transplantation is crucial regarding HSCT-related outcomes and long-term developmental effects in the youngest patients [11,12]. Patients undergoing allo-HSCT experience significant metabolic changes, with evidence suggesting a 30–50% increase in their basal metabolism and a systemic inflammatory syndrome is frequently activated [8]. Thus, nutritional support is essential in these patients and is considered a way to improve transplant outcomes in children [13,14,15,16,17]. Although recommendations for managing nutritional needs in pediatric oncology have recently been published [18], specific guidelines tailored to pediatric allo-HSCT recipients are unavailable, particularly given the complexity of addressing nutritional needs throughout different stages of childhood. However, considerable clinical studies in the literature have tackled this issue [19,20,21,22,23,24].

This study aims to evaluate if and how the BMI of pediatric allo-HSCT recipients influences transplant-related outcomes. Assessing BMI at set times along all patients’ therapeutic program, from diagnosis to the end of first-line treatment and the conditioning onset, we evaluated its potential correlation with the post-transplant course, including overall survival (OS), disease-free survival (DFS), transplant-related mortality (TRM), and event-free survival (EFS). Further, we tried to establish a BMI cut-off value to predict positive or negative transplant-related outcomes, such as early and late TRCs, and acute and chronic GVHD.

## 2. Materials and Methods

### 2.1. Study Design

This is a retrospective single-center observational study conducted at the Pediatric Bone Marrow Transplant Center of the Institute for Maternal and Child Health—IRCCS Burlo Garofolo, Trieste, Italy, from January 2003 to December 2023. The Institutional Review Board approved this study. This study was conducted according to the Declaration of Helsinki (reference No. IRB RC 44/24). Written informed consent for using any clinical data in research was obtained from parents or guardians.

The primary aim was to evaluate the impact of BMI assessed at the time of transplantation on OS and EFS. Secondary aims included the effect of BMI on engraftment and the occurrence of early and late TRCs. Additionally, we considered factors such as stem cell source, conditioning regimen, donor type, performance status, and total parenteral nutrition (TPN) concerning BMI assessed before allo-HSCT. We also investigated whether the BMI assessed when the primary disease is diagnosed and at the end of first-line treatment can influence the post-transplant course.

### 2.2. Study Population and Data Collection

From January 2003 to January 2023, 242 patients aged from 0 to 17 years, affected by hematological malignancies, non-malignant hematological diseases, and solid tumors, who underwent allo-HSCT, were included in this study. We included surviving patients with a minimum follow-up of 12 months. Inclusion criteria did not restrict donor type, graft source, primary disease, or conditioning regimen. Patients who were ≥18 years old at the time of transplantation, who underwent autologous transplantation, nonmyeloablative conditioning, and second or subsequent transplant attempt, were excluded from our study. Patients with missing or incomplete data were excluded from this study to avoid influencing the results.

The number of enrolled patients is calculated to obtain results with statistical power (10% dropout was considered acceptable to conduct this study). An intermediate effect size (f = 0.25) was estimated for the study endpoint. Enrolling more than 220 patients to detect a statistically significant association (*p*-value < 0.05) with a univariate analysis and a power of 90% was considered necessary. Sample size calculations have been performed using G*Power version 3.1.

The myeloablative protocols are defined as total body irradiation ≥8 Gy, busulfan 16 mg/kg, or melphalan 140 mg/m^2^ [25,26]. All patients were treated according to the standard myeloablative protocols based on chemotherapy and radiation dosing, as previously described [27]. GVHD prophylaxis was performed with a calcineurin inhibitor for the matched related donor (MRD), a calcineurin inhibitor, and mycophenolate mofetil for the matched unrelated donor (MUD), with the addition of post-transplant cyclophosphamide from 2013 in the case of a haploidentical donor [3,28]. Anti-thymocyte globulin was included in the myeloablative conditioning regimen with the MUD, haploidentical donors, and some hemoglobinopathies with the MRD. Prevention and treatment of infection and other transplant-specific supportive care were managed according to standard institutional practices. Patient characteristics, including age, gender, primary disease, severe adverse events, and outcomes related to primary treatments, were collected. Transplant-specific characteristics, such as Disease Risk Index (DRI), donor type, stem cell source, and conditioning regimen, were recorded as part of the patient’s medical history. We assessed DRI using defined criteria that included age, stage, cytogenetics, and disease status, including minimal residual disease at the time of allo-HCT [29,30]. Data on transplant-related outcomes were collected, including OS, EFS, TRM, early and late TRCs, recovery of neutrophils and platelets, acute and chronic GVHD incidence, and severity grading. OS was defined as the time from allo-HSCT until death from any cause, while TRM was defined as the time from allo-HSCT to death from any causes, including transplant-related complications and relapse. EFS was defined as the time from allo-HSCT to death or disease progression. OS and EFS were evaluated until 36 months. The relapse rate was calculated as the number of patients experiencing disease relapse relative to the total number of transplant recipients (affected by hematological malignancies or solid tumors). Both acute and chronic GVHD were evaluated according to standardized criteria based on a published staging system and clinical grading criteria [31]. As previously published, the term early TRCs included complications originating from localized or systemic endothelial damage, such as sinusoidal obstruction syndromes (SOSs), capillary leak syndromes (CLSs), engraftment syndromes (ESs), and transplant-associated thrombotic microangiopathies (TMAs) [32]. Late TRCs taken into consideration were growth and gonadal failure, thyroid dysfunction, avascular necrosis, osteoporosis, malignant complications, and other late events [33].

### 2.3. Assessment and Categorization of BMI in the Study Cohort

BMI data were collected at three time points: before and after first-line treatment (T1 and T2, respectively), and before transplantation, defined as T3.

BMI was calculated using the formula weight (kg)/height^2^ (m^2^). In pediatrics, BMI-for-age percentiles were classified using variable thresholds that define patients as normal, underweight, or overweight according to established criteria from the World Health Organization (patients aged <2 years) and the Centers for Disease Control and Prevention (from 2 to 20 years old). Variable thresholds derived from a reference population defined as a child growth reference consider differences associated with age and sex [34,35].

### 2.4. Parental Nutrition

In our Institute, TPN is provided by the hospital pharmacy, which prepares individual TPN bags for each patient according to their nutritional needs. These are assessed and modified daily based on laboratory analysis of blood samples. This standardizes care and ensures that each patient receives adequate macro- and micronutrients, even considering gender-based nutritional needs [36].

We collected each patient’s duration of TPN, the volume and number of TPN bags administered, the composition of the NPT bags, and the daily intake of calcium, phosphorus, and magnesium.

### 2.5. Statistical Analysis

We reported continuous variables using the median and interquartile range (IQR) for non-normally distributed variables. Qualitative variables were shown as frequencies and percentages. We used the chi-square or Fisher’s exact test to compare patients’ demographic and clinical characteristics for categorical variables. The Mann–Whitney rank-sum test was used to compare continuous variables. Receiver operating characteristic (ROC) curves were then constructed for BMI levels to determine the optimal cut-off, using the Youden index to predict the risk of TRC. Sensitivity and specificity values of the cut-off point were analyzed. The non-parametric Kendall tests examined the association between total kcal/Kg glucose kcal/Kg, fat kcal/Kg, NPT volume mL/Kg, and nutritional status. A multivariate analysis was performed to test the independence of the significant effects identified in univariate analyses on the phenotypes considered. For multivariate analysis, generalized linear models of the appropriate family were used, combining significant covariates in the univariate analysis as the independent variables.

All *p*-values < 0.05 were considered statistically significant. All statistical analyses were conducted using the software R, version 4.1.0.

## 3. Results

### 3.1. Patients

This study included 242 pediatric patients (160 males and 82 females) who underwent an allo-HSCT from 2003 until 2023. Detailed patient demographics and transplant characteristics are shown in Table 1. The entire cohort’s median age at HSCT was eight years (IQR, 4–13). Underlying diagnoses were grouped into acute lymphoblastic leukemia (36.8%), acute myeloid leukemia (20.7%), myelodysplastic syndrome (6.2%), non-malignant disease (31.4%), and solid tumor (4.9%). Interestingly, there is a correlation between T3 BMI values and primary disease groups. Figure 1 shows significant differences in BMI across five primary disease groups (*p* = 0.002). Patients diagnosed with solid tumors consistently demonstrated higher BMI (median (IQR): 19.45 (1.9)), and those with MDS had the lowest BMI at all time points (16.10 (1.9)), followed by non-malignant hematological disease, AML, and ALL (16.20 (1.9), 16.95 (3.72), and 17.90 (4), respectively).

### 3.2. Correlation Between Pre-Transplant BMI and Survival Outcomes

The three-year OS and EFS rates were 73.97% (*n* = 179) and 64.04% (*n* = 155), respectively. The OS was significantly different according to the T3 BMI. Underweight patients have a lower 12-month OS rate (Figure 2A) than overweight and normal-weight patients (46.81% vs. 75.44% vs. 89.13%, *p* = 1.22 × 10^−8^). OS at 36 months (Figure 2B) showed the same trend (42.55% vs. 73.68% vs. 84.78%, *p* = 8.88 × 10^−8^, respectively). Similarly, underweight patients experienced a notable decrease in EFS at 12 months compared to normal-weight and overweight patients (42.55% vs. 81.16% vs. 71.93%, *p* = 2.701 × 10^−6^, Figure 2C). This trend persists at 36 months (36.17% vs. 73.18% vs. 64.91%, *p* = 2.911 × 10^−5^, Figure 2D).

### 3.3. Correlation Between Pre-Transplant BMI and Other Transplant-Related Outcomes

DFS was 64.46% (*n* = 156). Forty-eight patients (19.8%) experienced primary disease recurrence after allo-HSCT. The overall mortality rate was 24.4% (*n* = 62), with an in-hospital mortality rate of 18.2% (*n* = 43). The causes of death were disease recurrence in 20 patients (33.9%), GVHD in 5 patients (8.5%), infectious complications in 15 patients (25.4%), and organ toxicity in 15 patients (25.4%). Furthermore, four patients (6.77%) died from causes unrelated to those previously mentioned.

Patients with high T3 BMI values (median (IQR): (17.6 (4.42) vs. 16.65 (4.07), *p* = 0.00068) have a significantly increased risk of aGVHD. Otherwise, T3 BMI was not significantly correlated with early TRCs and cGVHD (Figure 3).

### 3.4. Correlation Between Pre-Transplant BMI and Transplant-Related Variables

The correlation between the T3 BMI and transplant variables, including the source of stem cells, conditioning regimen, and donor type, is shown in Figure 4. Patients who received stem cells from bone marrow had a significantly lower T3 BMI compared to those who received peripheral blood stem cells (median (IQR): 16.75 (3.15) vs. 18.15 (5.64), respectively; *p* = 8 × 10^−4^, Figure 4A). Moreover, patients who received a myeloablative chemotherapy regimen exhibited a significantly lower BMI at HSCT compared to those who received a TBI regimen (16.45 (3.82) vs. 17.90 (4.77), respectively; *p* = 0.001, Figure 4B). A trend was observed when comparing the BMI of patients stratified by donor type (*p* = 0.052, Figure 4C). Patients who received stem cells from sibling donors had a higher median BMI than those who received the graft source from haploidentical or unrelated donors. Engraftment and DRI are not significantly correlated with T3 BMI (*p* > 0.05).

### 3.5. Impact of BMI at the Diagnosis and the End of Primary Treatment on Transplant Outcomes

Once we established that BMI significantly influences some HSCT outcomes, we assessed whether transplant outcomes could already be influenced by the time of primary diagnosis (T1) or during first-line treatment (T2).

We evaluated the correlations between the BMI at T1 and T2 and the transplant results. Underweight patients demonstrate a lower 12-month and 36-month survival rate than overweight and normal weight patients (Figure 5A, T1: 59.38% vs. 78.57% vs. 81.75%, respectively, *p* = 0.02441; T2: 50% vs. 79.69% vs. 84.29%, *p* = 3.592 × 10^−5^; Figure 5B, T1: 56.25% vs. 73.81% vs. 78.57%, *p* = 0.036; T2: 47.37% vs. 78.13% vs. 79.29%, *p* = 0.0002494). EFS rates were markedly lower in underweight patients compared to overweight and normal weight patients at both at 12 (Figure 5C) and 36 months (Figure 5D), at T1 (53.13% vs. 70.24% vs. 76.98%, *p* = 0.02698 and 46.88% vs. 58.33% vs. 72.22%, *p* = 0.01141, respectively, at 12 and 36 months) and T2 (50% vs. 71.88% vs. 77.14%, *p* = 0.004496 and 44.74% vs. 64.06% vs. 69.29%, *p* = 0.02002, respectively, at 12 and 36 months).

Patients with high BMI values recorded at T1 (median (IQR): 17.8 (5) vs. 16.35 (3.47), and T3 (17.6 (4.3) vs. 16.45 (3.65)) have a significantly increased risk of early TRCs (*p* = 0.016, *p* = 0.06, respectively Appendix A). BMI assessed at T2 (17.48 (4.59) vs. 16.32 (3.38)) was not significantly correlated with early TRC. Similarly, the occurrence of aGVHD (Appendix A) showed the same trend at T1 (18.4 (6.45) vs. 16.4 (3.52), *p* = 0.00051) and at T2 (18.15 (4.42) vs. 16.30 (3.77), *p* = 0.00033). Otherwise, the risk of cGVHD occurred in patients with low BMI at T1 (15.8 (1.40) vs. 17.4 (4.82), *p* = 0.006), at T2 (15.40 (2.61) vs. 17.25 (4.40), *p* = 0.034; Appendix A).

### 3.6. T1 and T2 BMI Distribution in Primary Diseases and Correlation with Primary Treatment Outcomes

The Kruskal–Wallis test revealed significant differences in BMI across the disease groups before (T1) and after (T2) primary treatment (T1: *p* = 0.004; T2: *p* = 0.005; Appendix A). Patients diagnosed with solid tumors consistently showed a higher BMI (median (IQR): (A) 20.15 (2.7); (B) 19.30 (2.5)), and those with MDS were the lowest at all time points ((A) 16.30 (2.7); (B) 15.90 (2.5)). The median BMI values for the different primary disease categories remain relatively consistent. In Appendix A, a significant difference in BMI was observed at T1 and T2 (T1, *p* = 6 × 10^−5^, T2: *p* = 0.00042) across the four groups of primary treatment outcomes. Patients who had experienced disease progression and were non-response exhibited a lower median BMI at T1 (16.20 (4.6) and 16.30 (2.7), respectively) compared to the complete responder and partial responder (18.45 (5.15) and 17.40 (4.6), respectively). Similarly, at T2, patients with lower BMI values were those who experienced disease progression or did not respond to treatment (16.20 (5.0) and 16.20 (2.3), respectively). The highest median BMI was measured in patients who achieved complete remission (18.25 (4.5)).

### 3.7. Analysis of the Correlation Between BMI and Other Variables

A positive correlation between BMI at different time points and the performance status at the time of allo-HSCT was shown in Appendix A (T1: r = 0.167, *p* = 7.4 × 10^−4^; T2: r = 0.168, *p* = 6.66 × 10^−4^; T3: r = 0.184, *p* = 1.87 × 10^−4^).

A proportion of 84.7% (*n* = 205) of patients received parenteral nutrition for a median of 15.5 (10.2–27.7) days. A significant inverse correlation was observed between BMI at different time points and NPT regarding total Kcal/Kg, glucose Kcal/Kg, fat Kcal/Kg, and volume/Kg (Appendix A). The protein Kcal/Kg correlation was not significant.

### 3.8. Multivariate Analysis

A multivariate analysis with a multiple linear regression model was performed, focusing on the variables found to be statistically significant in a univariate analysis. Table 2 shows the statistical significance of each variable. The potential confounding effect of the year of transplantation (from 2003 to 2023) on outcomes was initially assessed using a univariate approach. However, this effect was not significant in the multivariate analysis. Only primary treatment outcomes, graft source, donor type, and acute and chronic GVHD appear to impact BMI values assessed at diagnosis, at the end of primary treatment, and before transplant. In a multivariate analysis, primary diagnosis, HSCT conditioning regimen, early TRC, OS, and EFS were non-significant.

### 3.9. Evaluation of the BMI-for-Age Percentile Cut-Off Point and the Risk of TRC Onset

By exploring the relationship between nutritional status and post-transplant complications, we investigate the contribution of patients with the BMI cut-off value predisposed to or protected against TRCs. Receiver operating characteristic (ROC) curves were constructed to determine the optimal BMI-for-age percentile cut-off, using the Youden index, to evaluate the risk of transplant-related outcomes. As shown in Table 3, patients with a BMI below the cut-off had a lower probability of early TRC, including aGVHD, than those with a BMI above the threshold. In contrast, BMI values below the cut-off predict an increased risk of cGVHD.

## 4. Discussion

BMI is a commonly used anthropometric method to assess nutritional status and weight-related health risks. Its simplicity, non-invasiveness, and cost-free nature make it a valuable indicator and first-line screening tool. It determines weight categories and monitors patient changes over time, especially in response to treatments or interventions [37].

Our study collected BMI data at three distinct time points to provide a comprehensive view of the patient’s nutritional status throughout treatments. The first measurement was taken upon primary diagnosis, offering a baseline assessment of the patient’s condition before any therapeutic intervention. The second was recorded at the end of first-line treatment, which allowed us to evaluate any changes in BMI resulting from initial therapies, such as chemotherapy or radiation. The third measurement was taken before the transplantation procedure, providing a baseline for the patient’s nutritional state immediately before the conditioning regimen. By assessing BMI at these key stages, this study aimed to retrospectively identify potential correlations between nutritional status and clinical outcomes across the entire treatment period rather than focusing on the time of transplantation or the post-transplant course. BMI evaluation is already used in allo-HSCT. Studies in pediatrics and adults have identified the correlation between BMI before and after allo-HSCT procedures and transplant outcomes [24,38,39]. However, to our knowledge, there are no studies in pediatrics focused on nutritional status at diagnosis or the end of primary treatment correlating BMI at these times with post-transplant outcomes.

In this study, the BMI distribution across five primary disease groups underscores the heterogeneity in nutritional status among pediatric patients, with BMI variations potentially reflecting both the underlying disease and treatment-related factors. Notwithstanding the treatments and changes in the clinical condition of patients, the relative differences in BMI between disease groups remain consistent over time. Patients affected by MDS had the lowest BMI at all stages, which may reflect the impact of chronic hematologic disorders on growth and metabolism, more severe cachexia, and poorer nutritional status, which aligns with these findings [40]. The consistently low BMI in MDS patients suggests that they are particularly vulnerable to malnutrition throughout their treatment. High BMIs in patients affected by solid tumors might reflect better nutritional status or less severe disease-related cachexia compared to other disease groups. This is consistent with the literature suggesting that solid tumors might not be associated with the same degree of weight loss or muscle wasting as in hematologic malignancies [41].

The analysis of BMI values across time and primary treatment outcomes reveals significant differences at both diagnosis and the end of treatment, highlighting the impact of treatment responses on nutritional status [42,43]. A lower BMI before and after primary therapy was observed in non-responder or relapsed patients. A low BMI could reflect the underlying severity of their condition or a more advanced stage of the disease, which could contribute to nutritional deficits and poorer overall health status. The optimal stem cell source for patients with a higher BMI is PBSCs harvested by leukapheresis after G-CSF donor mobilization rather than the iliac crest bone marrow aspiration. This approach is preferred because a higher BMI requires harvesting the greatest number of stem cells due to the greatest kilogram number. This approach is preferred because patients with a higher BMI require the greatest number of stem cells due to the greatest kilogram number, making the bone marrow collection procedure very long and expensive for the collection facility and unfavored by donors. In these cases, the PBSC collection is the most chosen option. The choice of conditioning regimen administered before the stem cell infusion is determined by factors related to the patient’s characteristics and primary disease [44]. Typically, patients receiving TBI conditioning have lymphoblastic leukemia as their primary disease. It is generally treated with less intensive frontline therapy compared to myeloid, allowing patients to recover hematopoiesis between treatment cycles quickly [45,46]. Consequently, patients undergoing TBI tend to have a higher BMI than those receiving chemotherapy. In contrast, myeloid treatment regimens are more aggressive, with shorter intervals between cycles, leading to more severe aplasia and a higher risk of malnutrition. This accounts for the lower BMI observed in patients than those treated with TBI.

This study has shown that being underweight before and after primary treatment and before allo-HSCT is associated with a poorer OS and EFS in pediatric patients. The observed data are consistent with the literature, as patients with a lower BMI are known to have a lower tolerance to the conditioning regimen, resulting in a higher mortality rate. In addition, a lower BMI may reflect the presence of a more aggressive disease that requires more aggressive primary treatments and causes a greater degree of physiologic imbalance [39]. Other published data have shown that both underweight and overweight are associated with worse outcomes in children with ALL and AML [45,46].

Nutritional status is a potentially modifiable risk factor for improving EFS and morbidity in pediatric patients. Patients with a low BMI are often severely malnourished due to chronic disease, gastrointestinal disorders, or an inability to absorb nutrients effectively. NPT is an important intervention for these patients to prevent further weight loss and provide the necessary calories, protein, and micronutrients they cannot obtain through oral or enteral nutrition. This results in greater NPT use in subjects with a lower BMI, which is consistent with our result [47].

The impact of overweight on outcomes after allo-HSCT remains debated. However, it is important to note that results from studies on overweight or underweight in adults may not be directly applicable to children. Meta-analyses show that overweight patients often have poorer outcomes compared to non-overweight patients [39]. Ideal body weight might better predict cell dose adequacy than actual body weight, and excessive stem cell dosing could increase the risk of aGVHD. Moreover, overweight patients might experience altered drug pharmacokinetics affecting GVHD prophylaxis. In addition, adipocytokines like leptin, which are elevated in those with higher fat mass, can influence T-regulatory cell function and aGVHD occurrence. Our findings align with existing research: higher BMI values are associated with an increased risk of early TRCs and aGVHD, while lower BMI values correlate with a higher risk of cGVHD [48,49]. The timing of BMI measurement is crucial. Baseline and pre-transplant BMI are more predictive of TRCs and aGVHD. The identified age-specific BMI percentile cut-offs accurately predicted the risk of early TRCs, including aGVHD and cGVHD. Adhering to these cut-offs enables clinicians to reduce the likelihood of complications, thereby improving clinical outcomes and enhancing patients’ quality of life. Moreover, Koo et al. (2023) show that higher BMI in children is associated with chronic inflammation and endothelial dysfunction, particularly transplantation-associated thrombotic microangiopathy. This identifies a high-risk group that could benefit from early interventions to prevent or address endothelial damage [50]. In vivo studies in mice have also revealed a possible role of obesity-related changes in the gut microbiota and its role in the occurrence of intestinal aGVHD [51]. Modifying the microbiota composition could be a promising strategy to prevent or treat this post-transplant complication. Antibiotic stewardship in transplantation may help maintain the native microbiota and improve immune responses [52].

BMI has several limitations, leading to low sensitivity in diagnosing overweight and obese patients. For example, individuals with a high body fat percentage (e.g., >30%) are often classified as within the normal weight range based on BMI alone. To overcome this, other methods that assess body composition can provide more accurate evaluations of fat mass, lean mass, and overall body composition. These techniques offer a more comprehensive view of an individual’s health status than BMI alone. Arm anthropometry, for instance, is a simple and cost-effective way to estimate body composition, though it only approximates muscle mass. More precise measurements can be obtained through dual-energy X-ray absorptiometry (DEXA), but this method is expensive and unavailable everywhere. However, bioelectrical impedance analysis (BIA), a more affordable and portable option, provides reasonably accurate body composition measurements compared to DEXA, making it a practical choice for adults and children without radiation [24,53].

One limitation of our study is that we did not analyze body composition about lean mass and adipose tissue. Further investigation into how body composition changes before and after transplantation, particularly about treatments and outcomes, would undoubtedly prove valuable. For example, Feng et al. (2018) demonstrate that body composition in children with aGVHD changed significantly during the first 100 days following HSCT, with a marked decrease in lean body mass and an increase in adipose tissue [53]. Hartog et al. (2024) conducted a retrospective analysis of physical performance 100 days after HSCT in pediatric patients. Patients exhibited lower hip flexor muscle strength and appendicular skeletal muscle mass, along with a slower increase from floor time 100 days after HSCT compared to the average values of the normal population. Muscle strength was assessed by hand-held dynamometry, and the muscle mass was estimated using BIA; the study included a physical functioning assessment [54].

Future studies could benefit from incorporating body composition assessments before and after transplantation to understand better the impact of allo-HSCT and transplant-related complications on patient outcomes.

## 5. Conclusions

These findings underscore the importance of nutritional assessment and management in the pediatric population undergoing allo-HSCT, as deviations from normal BMI can significantly impact both short-term outcomes and long-term post-transplant health. The results of our study could help identify patients at higher risk of early complications and define more effective nutritional management strategies before and after transplantation.

## Figures and Tables

**Figure 1 nutrients-16-03638-f001:**
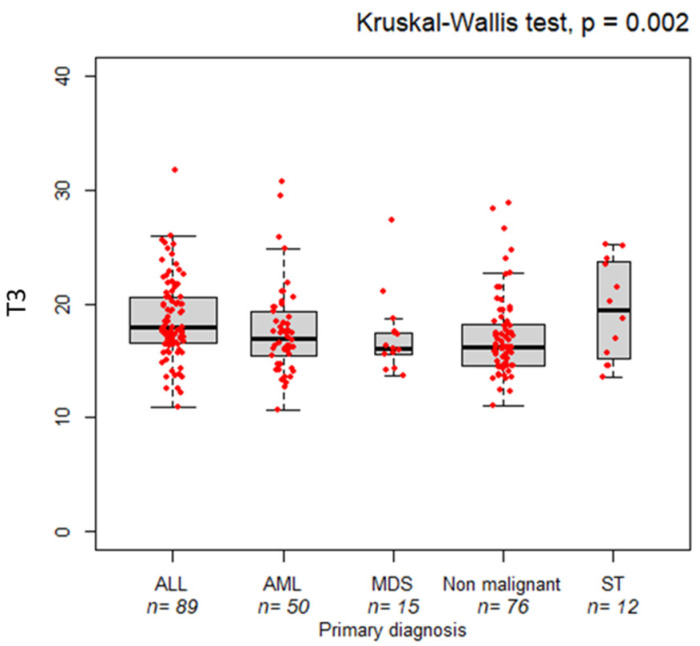
Correlation between BMI and primary disease. Boxplots compare the pre-transplant BMI (T3), across five primary disease groups. The bold horizontal line represents the median value. ALL: acute lymphoblastic leukemia, AML: acute myeloid leukemia (AML); MDS: myelodysplastic syndrome (MDS); ST: solid tumor. *p*-values are from the Kruskal–Wallis test.

**Figure 2 nutrients-16-03638-f002:**
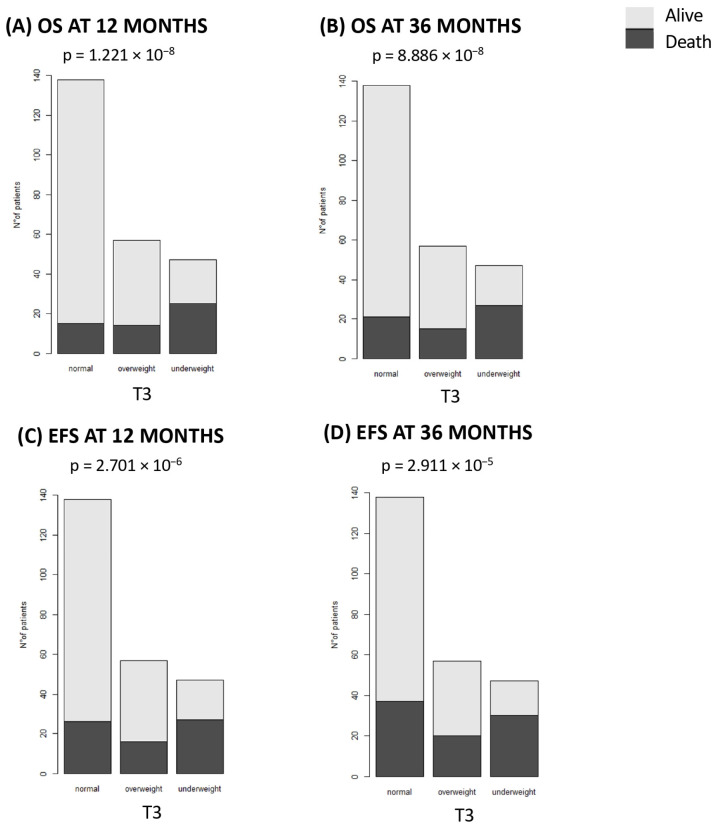
Overall survival (OS) and event-free survival (EFS) at 12 and 36 months. The figure illustrates the survival differences across BMI categories before allo-HSCT (T3). (**A**) OS at 12-months; (**B**) OS at 36-months; (**C**) EFS at 12-months; (**D**) EFS at 36-months.

**Figure 3 nutrients-16-03638-f003:**
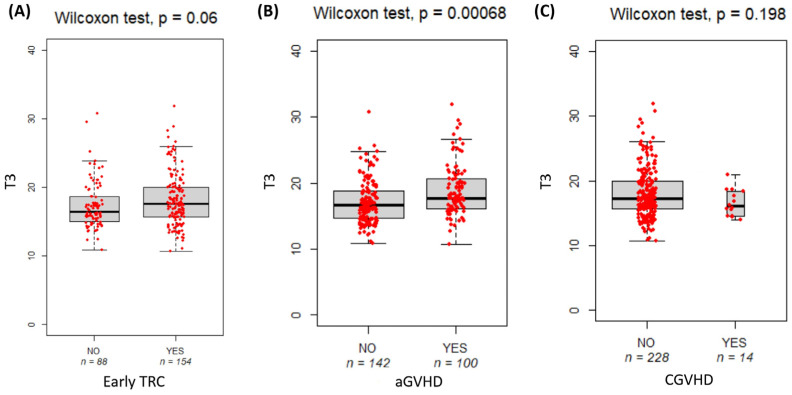
Boxplot compares pre-transplant BMI (T3) with the risk of early TRCs (**A**), acute GVHD (**B**), and chronic GVHD (**C**). The bold horizontal line represents the median value. *p*-values are from the Wilcoxon test. GVHD: graft-versus-host disease.

**Figure 4 nutrients-16-03638-f004:**
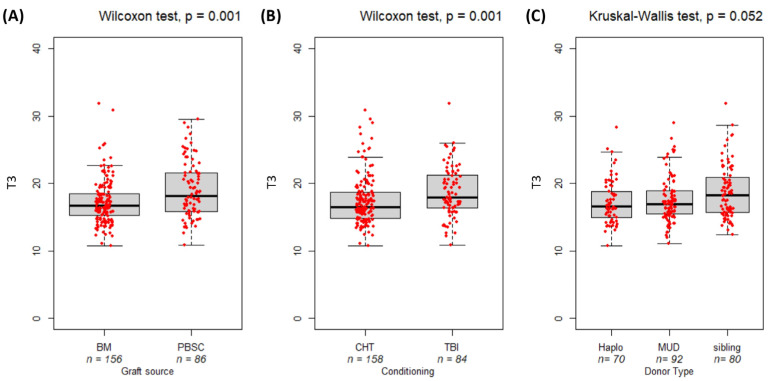
Boxplot comparing pre-transplant BMI (T3) and transplant-related variables: Panel (**A**) graft source (BM: bone marrow, PBSC: peripheral blood stem cells), Panel (**B**) conditioning (CHT: chemotherapy; TBI: total body irradiation), Panel (**C**) donor type (Haplo: haploidentical donor; MUD: matched unrelated donor). The bold horizontal line represents the median value. *p*-values are from the Wilcoxon test (**A**,**B**) and the Kruskal–Wallis test (**C**).

**Figure 5 nutrients-16-03638-f005:**
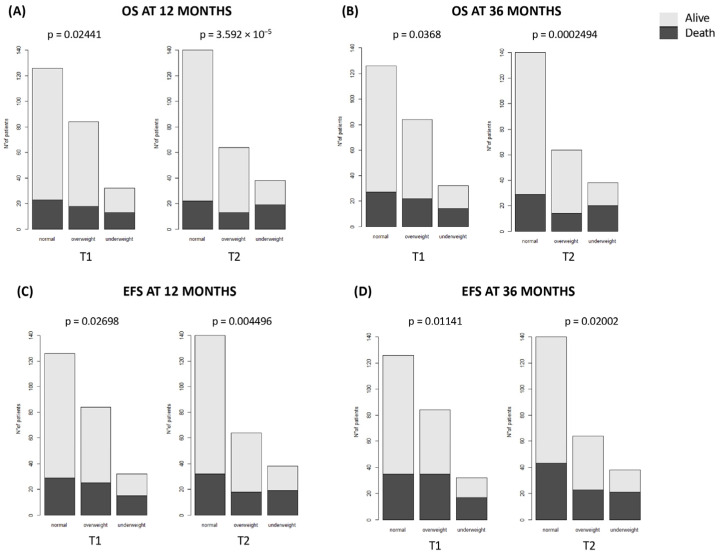
Overall survival (OS) and Event-Free Survival (EFS) at 12 and 36 months. The figure illustrates the survival differences across BMI categories before (T1) and after first-line treatment (T2). (**A**) OS at 12-months; (**B**) OS at 36-months; (**C**) EFS at 12-months; (**D**) EFS at 36-months.

**Table 1 nutrients-16-03638-t001:** Demographics and clinical characteristics of allo-HSCT recipients. MCHT, myeloablative chemotherapy; TBI, total body irradiation; DRI, Disease Risk Index; IQR, interquartile range.

Demographics and Clinical Data	Cohort (*n* = 242)
**Age at transplant** Median (IQR), years	8 (4–13)
**Gender, number (%)** Male, *n* (%) Female, *n* (%)	160 (66%) 82 (34%)
**Primary disease, number (%)** Acute lymphoblastic leukemia (ALL) Acute myeloid leukemia (AML) Myelodysplastic syndrome (MDS) Solid tumor Non-malignant disease	89 (36.8%) 50 (20.7%) 15 (6.2%) 76 (31.4%) 12 (4.9%)
**DRI, number (%)** Low risk Medium risk High/Very high risk	108 (44.6%) 72 (29.8%) 62 (25.6%)
**Lansky Performance Status ** Median (IQR)	100 (80–100)
**Allogenic donor type, number (%)** Matched unrelated donor Matched related donor/sibling donor Haploidentical donor	92 (38%) 80 (33%) 70 (29%)
**Stem cell source, number (%)** Bone marrow Peripheral blood stem cell (PBSC)	156 (64.5%) 86 (35.5%)
**Myeloablative conditioning regimen, number (%)** MCHT-based TBI-based	158 (65.3%) 84 (34.7%)
**BMI at diagnosis, number (%)** Normal Underweight Overweight	126 (52.1%) 32 (13.2%) 84 (34.7%)
**BMI at the end of primary treatment, number (%)** Normal Underweight Overweight	140 (57.8%) 38 (15.7%) 64 (26.4%)
**BMI at transplant, number (%)** Normal Underweight Overweight	138 (7%) 47 (19.4%) 57 (23.5%)

**Table 2 nutrients-16-03638-t002:** Multivariate analysis between BMI levels and clinical parameters. BMI was assessed before (T1) and after (T2) first-line treatment and pre-transplant (T3). aGVHD: acute graft-versus-host disease; cGVHD: chronic graft-versus-host disease. Ns: not significant.

Parameter	T1 BMI	T2 BMI	T3 BMI
**Years of allo-HSCT**	Ns	Ns	Ns
**Graft source**	PBSC: 2.02 × 10^−6^	PBSC: 3.75 × 10^−6^	PBSC: 1.8 × 10^−5^
**Donor type**	Sibling: 0.021	Sibling: 0.044	Ns
**aGVHD**	0.025	0.0076	0.049
**cGVHD**	0.002	0.0023	0.0198

**Table 3 nutrients-16-03638-t003:** AUC analysis to determine the optimal BMI-for-age percentile cut-off for transplant-related outcomes. AUC: area under the curve; OR: odds ratio; CI: confidence interval; TRC: transplant-related complications. aGVHD: acute graft-versus-host disease; cGVHD: chronic graft-versus-host disease. Ns: not significant; NA: not applicable.

Transplant-Related Outcomes	BMI Time Point	AUC	Cut-Off (BMI-for-Age Percentiles)	OR (CI 95%)	Sensitivity (%)	Specificity (%)	*p*-Value	Below Threshold Effect
Early TRC	T1	0.57	82.5	0.39 (0.21–0.72)	40.91%	78.41%	0.0022	↓ risk
	T2	0.55	81.5	0.53 (0.29–0.96)	37.01%	76.14%	0.035	↓ risk
	T3	0.55	36.0	0.62 (0.36–1.06)	66.88%	44.32%	Ns (0.083)	NA
aGVHD	T1	0.62	82.5	0.37 (0.21–0.64)	47%	75.35%	0.00029	↓ risk
	T2	0.63	22.5	0.31 (0.16–0.60)	86%	34.51%	0.00034	↓ risk
	T3	0.61	38.5	0.35 (0.20–0.62)	76%	47.18%	0.00025	↓ risk
cGVHD	T1	0.73	77.5	8.54 (1 × 10^−10^–1 × 10^10^)	100%	38.6%	0.0036	↑ risk
	T2	0.68	31.0	6.55 (1.98–21.64)	71.43%	72.37%	0.00053	↑ risk
	T3	0.63	52.5	3.73 (1.01–13.73)	78.57%	50.43%	0.035	↑ risk

## Data Availability

Data will be made available on request. The original contributions presented in this manuscript are included in the article. Further inquiries can be directed to the corresponding author.

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
