# Peer review of "Impact of Body Mass Index on Outcomes in Pediatric Allogeneic Hematopoietic Stem Cell Transplantation Recipients: A Single-Center Retrospective Study"

_nutrients, 2024, doi:10.3390/nu16213638_

Round 1

Reviewer 1 Report

Comments and Suggestions for Authors

While host factors such as performance status, organ function and nutritional status, which are the focus of the current paper, are important for allogeneic transplant outcomes, primary disease factors such as the nature and progression of the underlying disease, histrogical grade, treatment response and cytogenetic characteristics are also important as determinants.

The study included a highly heterogeneous patient population and did not provide sufficient information other than nutritional status to influence transplant outcomes, making it difficult to properly and scientifically assess the impact of BMI on patient survival.

It would be better to exclude at least cases in which the primary disease has relapsed from the analysis.

Author Response

While host factors such as performance status, organ function and nutritional status, which are the focus of the current paper, are important for allogeneic transplant outcomes, primary disease factors such as the nature and progression of the underlying disease, histrogical grade, treatment response and cytogenetic characteristics are also important as determinants.

The study included a highly heterogeneous patient population and did not provide sufficient information other than nutritional status to influence transplant outcomes, making it difficult to properly and scientifically assess the impact of BMI on patient survival.

It would be better to exclude at least cases in which the primary disease has relapsed from the analysis.

Based on the reviewer's suggestions, we have excluded relapsed patients from the analysis. The statistical analysis did not reveal any differences in OS, EFS, and TRC (P-value > 0.05) compared to the previous findings in the manuscript. The significance remains consistent with the original trend, so we deemed it unnecessary to replace the analyses. Therefore, despite the heterogeneous population noted by the reviewer, this does not affect the overall statistical analysis of the entire cohort.

Reviewer 2 Report

Comments and Suggestions for Authors

This is my review on «Impact of Body Mass Index on Outcomes in Pediatric Allogeneic Hematopoietic Stem Cell Transplantation Recipients: a Single-Center Retrospective Study".

The study is well designed. Authors should include a power analysis for the chosen sample size.

The study should include more information on how missing or incomplete data was handled. Were patients with incomplete records excluded?

BMI has several limitations, particularly in children as it does not distinguish between muscle mass and fat mass, so authors should mention it in limitations and suggest future studies with extra methods.

The results are presented well. Authors should describe why certain variables were included in the multivariate models and how they were chosen (was there any clinical relevance, etc.). Moreover, authors could include a few subgroup analyses.

Discussion is interesting. You should also analyze how nutritional interventions before and after transplantation could improve outcomes.

Figures and tables are fine.

Author Response

This is my review on «Impact of Body Mass Index on Outcomes in Pediatric Allogeneic Hematopoietic Stem Cell Transplantation Recipients: a Single-Center Retrospective Study".

The study is well designed. Authors should include a power analysis for the chosen sample size.

We thank the reviewer for his/her comments.

In the text, we added the power analysis for the chosen sample size as suggested in lines 150-154:The number of enrolled patients is calculated to obtain results with statistical power (10% dropout is acceptable to conduct the study). For the study endpoint, an intermediate effect size (f = 0.25) is estimated; it will be, therefore, necessary to enroll more than 220 patients to detect a statistically significant association (p-value < 0.05) with univariate analysis and with a power of 90%. Sample size calculations have been performed using G*Power version 3.1”.

The study should include more information on how missing or incomplete data was handled. Were patients with incomplete records excluded?

We apologize for the lack of clarity in the Methods section. To avoid influencing the results, we excluded patients with missing or incomplete data from the analysis. Accordingly, we have provided this clarification in the manuscript, Material and Methods section, at lines 148-149.

BMI has several limitations, particularly in children as it does not distinguish between muscle mass and fat mass, so authors should mention it in limitations and suggest future studies with extra methods.

Thank you for pointing this out. We agree with this comment. In lines 523-539, we mention the limitations of the study. Future studies could benefit from incorporating body composition assessments, including arm muscle area, BIA, or DEXA, to better understand body composition, focusing on lean mass and adipose tissue before and after allo-HSCT and transplant-related complications on patient outcomes.

The results are presented well. Authors should describe why certain variables were included in the multivariate models and how they were chosen (was there any clinical relevance, etc.). Moreover, authors could include a few subgroup analyses.

We thank the reviewer for his/her comments. Multivariate analysis was conducted using a multiple linear regression model, focusing on the variables found to be statistically significant in univariate analysis.

The authors categorized BMI into three subgroups: BMI assessed before and after first-line treatment (T1 and T2, respectively) and before transplantation, defined as T3 (Materials and methods section, lines 188-197). Specifically, cohort data was analyzed retrospectively to determine the correlation between BMI and transplant-related variables or outcomes.

Discussion is interesting. You should also analyze how nutritional interventions before and after transplantation could improve outcomes.

We thank the reviewer for his/her comments. We agree that nutritional interventions before and after transplantation can play a crucial role in improving outcomes. Pre-transplant nutrition can help optimize patient health, potentially reducing complications during and after the procedure. After transplantation, targeted nutritional support can enhance recovery and support long-term immune health. Although recommendations for managing nutritional needs in pediatric oncology have recently been published [Reference 18], specific guidelines tailored to pediatric allo-HSCT recipients are unavailable, particularly given the complexity of addressing nutritional needs throughout different stages of childhood. However, considerable clinical studies in the literature have tackled this issue [References 19-24].

Parenteral nutrition was administered for a median of 15.5 (10.2-27.7) days to 84.7% (n=205) of patients. Our department avoids the use of pre-formulated commercial TPN bags. Commercial bags follow a standard composition designed to meet general nutritional needs but may not meet the specific needs of a particular patient. However, it should be emphasized that there is a close collaboration between pharmacists, nutritionists, and clinicians.          
The pharmacy at IRCCS Burlo Garofolo (Trieste, Italy) prepares individual bags for total parental nutrition for each patient, according to individual nutritional needs, customized and modified daily based on laboratory analyses of blood samples. Each bag is tailored to meet the patient's specific nutritional needs based on their medical condition, gender, and requirements. This standardizes care and ensures that each patient receives adequate macro- and micronutrients. (Material and methods section at lines 199-206; Results section at lines 363-367).  

Figures and tables are fine.

We appreciate the reviewer's consideration.